# FROM APPEARANCE TO MOTION: ALIGNING VISUAL REPRESENTATIONS FOR ROBOTIC MANIPULATION

## ABSTRACT

Pre-trained vision models used in robotics often misalign with manipulation tasks due to the loss used to train these vision models being focused on appearance rather than motion. In order to enhance motion encoding within vision models, we introduce a simple novel contrastive training framework that operates over predictions of motion. After training over EPIC Kitchens, model evaluations on behavioral cloning show a improvement in success rate over state-of-the-art methods across a benchmark of 3 environments and 21 object manipulation tasks.

## 1 INTRODUCTION

Vision models used in robotics often derive from those trained for detection in still images or less commonly, in video clips. This practice can result in a misalignment between the robot's objective — effective manipulation (motion) — and the visual model's objective (appearance[1]). For robotics, especially in manipulation tasks, it is crucial to model motion since manipulation is fundamentally defined by motion rather than appearance.

Video representation learning often overfits to characteristics of action besides movement, such as gross-level scene and object appearance (Sevilla-Lara et al., 2021). This overfitting limits the utility of these models in robotic applications where motion is a critical factor. In order for robotic applications to be more effective it is important for visual models to capture motion characteristics, as this incorporation allows a better alignment between the robot's goals of effective manipulation and the capabilities of the visual model.

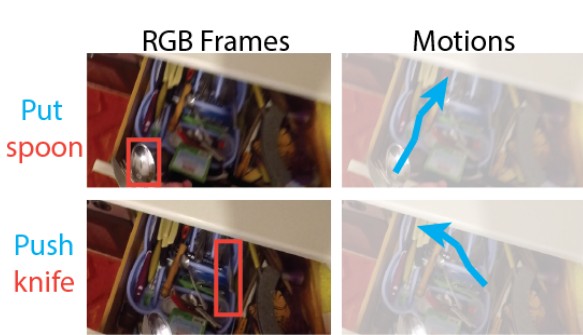

Figure 1: Illustration of the distinction between: a) appearance representations (RGB Frames) and, b) motion representations, for two verbs and two noun categories. Actions are difficult to distinguish on the basis of appearance alone, but the distinction becomes immediately apparent from the motions of the action.

Consider Figure 1, which illustrates two actions—placing and pushing—that are visually similar but distinct in their motions. While the visual appearance of these actions may be almost identical, the motions involved distinguish them. The difference between placing and pushing lies in their motion characteristics rather than their appearance characteristics.

Constraining a vision network to focus on motion is readily achieved via a contrastive loss. We introduce in this paper such a contrastive loss. Our contrastive learning approach utilizes motion as a self-supervised cue. This approach forces similar motions to be grouped together and dissimilar motions to be distinguished, organizing visual representations based on motion similarity. Consequently, any appearance characteristic that does not contribute to motion understanding is deemed irrelevant, ensuring that the network's focus remains on capturing the dynamics of motion. This

---

[1]From here on out we use "appearance" to refer to characteristics which are perceivable in static images such as color, shape, texture, spatial arrangement, etc.

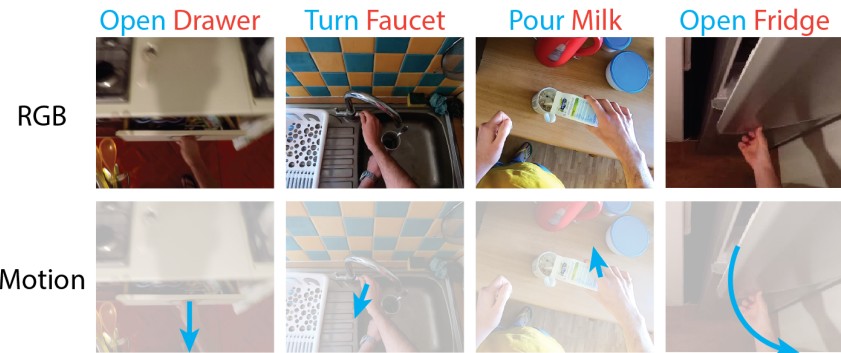

Figure 2: Example actions as shown in the RGB frames, and the direction of movement projected over the 2D plane of the image. Action categories are not immediately apparent from their single frame RGB representations alone; the motion characteristics make the action category clear.

targeted approach enhances the network's capability to learn and generalize from motion data effectively. This approach is made feasible by recent advances in motion segmentation, and eliminates the need for difficult and time-consuming ground truth collection, particularly for video data where action labels are challenging to define.

We evaluate our learned vision representations in the domain of robot motor policy learning, a domain which benefits from the introduction of motion centered representations. It has been demonstrated empirically (Nair et al., 2022; Hu et al., 2023) that improvements observed in simulation over pre-trained vision networks translate to improved success rate in the real world. Our method outperforms (measured by success rate) alternative approaches (Nair et al., 2022) that rely on temporal action boundaries or language as supervisory cues, which are less effective than our method at emphasizing motion characteristics and suffer from scalability issues due to the annotation burden.

Video datasets can come from a variety of sources, including those of human actions. Large-scale video datasets of human interaction can be collected more easily than robot manipulation datasets, which require aggregating videos across different robots in controlled settings. We choose the egocentric perspective, as they are especially useful in capturing hands and objects in interaction, offering a rich, diverse, multi-cue sensory stream. Self-produced motion is commonly in full view from the egocentric perspective, making motion particularly important to model. See Figure 2 for examples. As such, we adopt the EPIC Kitchens dataset (Damen et al., 2018).

To summarize: vision models used in robotics often misalign with manipulation tasks because they are trained for appearance-based objectives. By employing a contrastive loss to emphasize motion, our approach aligns visual representations with robotic objectives.

Our primary contributions can be summarized as followed:

- We observe that visual encoders commonly used in robotic control policies are not trained in such a way as to model the internal motion dynamics of actions. Robot control policies can benefit from visual representations modeling motion. We introduce a novel contrastive training framework which explicitly enhances motion encoding in the visual representation network.

- Qualitatively, our contrastive training framework improves the visual representation network's sensitivity to motions and its robustness to appearance change.

- Quantitatively, we evaluate our contrastive training framework on three behavior cloning RL environments. The experiments show that the visual representation network overall yields an improvement over state-of-the-art methods across a benchmark of 3 environments and 21 object manipulation tasks.

## 2 RELATED WORK

### 2.1 LEARNING VISUAL REPRESENTATIONS FOR MOTOR CONTROL

Large-scale pre-training Mahajan et al. (2018); Sun et al. (2017) over in-the-wild video datasets Damen et al. (2018); Grauman et al. (2022); Li et al. (2015) has long been considered a promising direction for learning useful visual representations for robotics applications. The pre-training paradigm has been explored in the domain of policy learning for motor control Shridhar et al. (2022); Nair et al. (2022); Yuan et al. (2022a); Hu et al. (2023), where visual representations are frozen during the training of motor policies.

There have been many recent works exploring the learning of visual representations over simulated robotic action Yen-Chen et al. (2020); Bousmalis et al. (2023); Yuan et al. (2022b) (but simulated environments suffer from lack of diversity), large-scale image data Dasari et al. (2023) (which result in misalignment between vision and policy objectives), small-scale real-world robot action videos Radosavovic et al. (2023) (which are difficult to collect), and large-scale human action videos Nair et al. (2022); Xiao et al. (2022). In this work, we pre-train a network over a large-scale human action video dataset - the EPIC Kitchens dataset Damen et al. (2018).

### 2.2 SELF-SUPERVISION OVER VIDEO

There have been a plethora of works which align video representations with language targets for learning useful representations for downstream language-conditioned robotics tasks Mees et al. (2022); Lynch & Sermanet (2020); Ma et al. (2023) and for robotic manipulation tasks conditioned on vision alone Nair et al. (2022); Sontakke et al. (2024). In an attempt to learn video representations with less explicit supervision, there has been significant attention exploring how time can be used as a self-supervisory signal. The majority of existing works fall within two categories: 1) approaches that enforce the feature-level distance between different video representations to be proportional to their respective distances over time Tanwani et al. (2020); Zhou et al. (2021); Nair et al. (2022), and 2) approaches that predict visual states in the future using a learned video dynamics model Wu et al. (2020); Manuelli et al. (2020). While these approaches implicitly learn the motions, the explicit adoption of motion as a form of self-supervision has only been applied over the field of action understanding Huang et al. (2021); Dessalene et al. (2024). To our knowledge, there is no existing work exploring 3D motion as a supervisory cue in the learning of frozen representations for robotics.

## 3 METHODS

Our goal is to leverage large, diverse, and naturalistic video datasets of human interaction as a substitute for real-world robot data that is 1) difficult to collect at scale, 2) only collected for specific embodiments, and 3) difficult to acquire for complex manipulation tasks. We learn a visual representation through a contrastive learning formulation detailed in Section 3.1. This involves vision architectures detailed in Section 3.2, and is trained through use of an egocentric RGB dataset detailed in Section 3.3. We then deploy the visual representations learned over these videos for learning motor control policies.

### 3.1 CONTRASTIVE LEARNING FORMULATION

We assume access to a dataset of RGB video clips $v \in V$ encompassing object manipulation actions. For each frame $f_k$ in video clip $v$ with length $t$ frames, there exist objects $o_{k_i} \in O_k$, each paired with a homogeneous transformation $T_{k_i}$ (with respect to the origin at the center of the camera) for each frame $k$ in $\{1, 2, .., t\}$. Our approach predicts 3D motion pertaining to object $o_{k_i}$ from RGB video, making our method independent of linguistic annotations or clip boundary annotations, unlike other approaches.

Our contrastive learning approach operates over the aforementioned frames $f_k$ and 3D relative motion, as shown in Figure 3. In formulating the training loss we randomly sample two frames $f_n$ and $f_m$ from video clip $v$, extracting frame-level feature maps using a backbone encoder. We extract image features $q_n$ and $q_m$, encouraging the feature representations to be aware of the motions acting upon them by predicting the motions from these features. Specifically, we compute the 3D relative

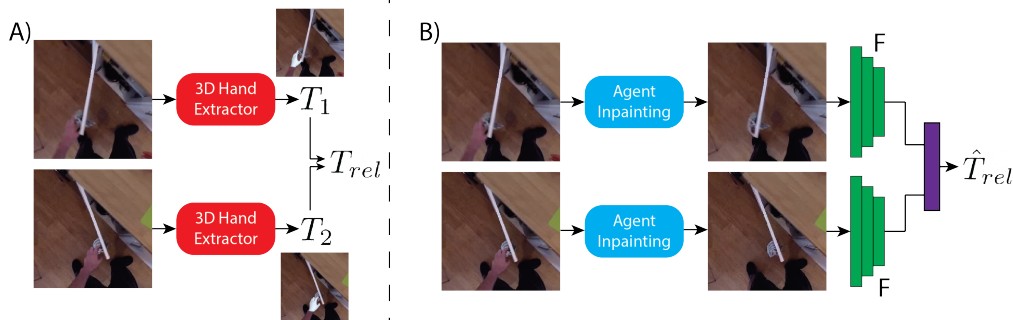

Figure 3: Illustration of our contrastive learning approach for leveraging video datasets in learning useful frozen representations for robot policy learning. From the same clip, we sample 2 frames for which hands and objects are in contact. A) We first infer 3D relative hand transformation $T_{rel}$ by feeding both frames to a 3D hand extractor (Pavlakos et al., 2024) and computing the difference in 3D hand predictions. B) We then feed both frames to an agent inpainting network (Chang et al., 2024), removing the hands from each image. Both inpainted frames are processed through a vision backbone (F), over which the relative transform between object states is predicted. This enables the model to learn representations that are sensitive to object motion.

motion $T_{rel}$, where $T_{rel}$ is tied to the relative motion of object $o_{m_i}$ from frame $m$ to $n$. We predict $\hat{T}_{rel}$ using a fully connected head over image features $q_n$ and $q_m$, supervising the head with mean squared error (MSE).

## 3.2 VISION MODELS

For consistency with the collection of models benchmarked in (Hu et al., 2023), our backbone network of choice is the ResNet50 network (He et al., 2016). In practice, the backbone of choice can be any network that preserves spatial locality.

## 3.3 DATASET

The following are desirable properties for a dataset to be used in our formulation: 1) The dataset features videos of everyday object manipulation, similar in distribution as to be encountered by the robot; and, 2) the data contains rigid transformation tracking ($T_{k_i}$) for one or more objects of interest; and, 3) the data does not lead to trained networks that are over-reliant on the embodiment of the (human) actor and therefore does not generalize to images containing the embodiment of a robot. The EPIC Kitchens dataset satisfies criterion 1).

Regarding criterion 2), as the data does not feature rigid transformation tracking ($T_{k_i}$) for the objects of interaction, we deploy a 3D hand model predictor (Pavlakos et al., 2024) as a substitute for the 3D pose of the objects of interaction. This exploits the fact that object motion and hand motion are strongly correlated during periods of contact between the two. We extract hands in 3D for frames in which contact is predicted between the hand and the environment using an off-the-shelf hand-object contact detector (Shan et al., 2020). As such, we calculate relative object transform $T_{rel}$ as the difference in predicted 3D hand vertices between frames $m$ and $n$, where $T_{rel}$ is of dimensionality $27 \times 3$. By representing $T_{rel}$ as 27 points belonging to the hand in contact with the object, $T_{rel}$ captures aspects of object motion.

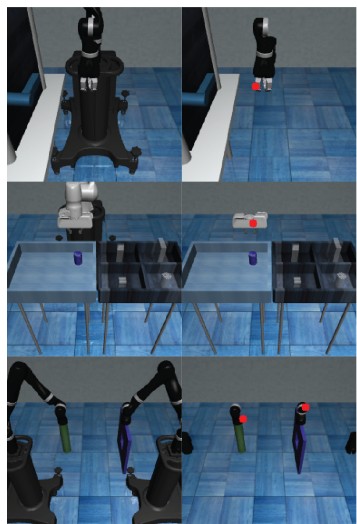

Figure 4: RGB frames where each row corresponds to a different task. Left column corresponds to images taken from the original environment, and right column corresponds to images after the robot body has been removed (the red circle is used for an ablation, see Section 4.3).

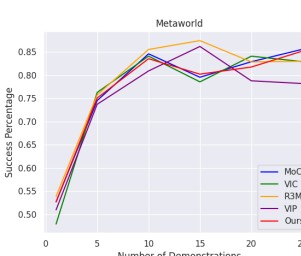 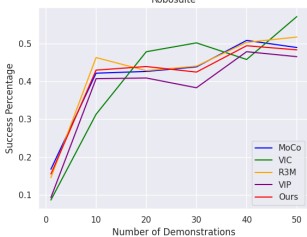 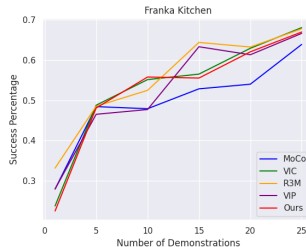

Figure 5: Behavioral cloning success rates (as Inter-Quartile Median (IQM)) for our model along with 4 SOTA benchmarks (VIP Ma et al. (2022), (Chen et al., 2020), VICReg (Bardes et al., 2021), and R3M (Nair et al., 2022)) over number of demonstrations for all three environments (MetaWorld, RoboSuite, and Franka Kitchen). These experiments are conducted over the original demonstrations, without any removal of the robot body in either the training demonstrations or simulation environment.

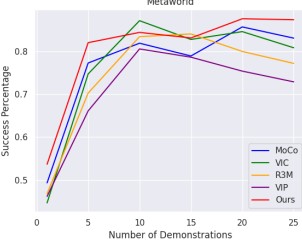 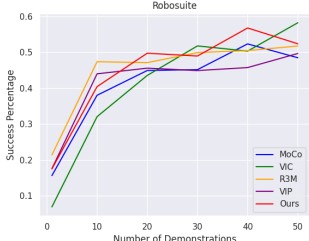 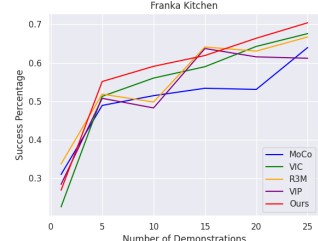

Figure 6: Behavioral cloning success rates (as Inter-Quartile Median (IQM)) for our model along with 4 SOTA benchmarks (VIP Ma et al. (2022), (Chen et al., 2020), VICReg (Bardes et al., 2021), and R3M (Nair et al., 2022)) over number of demonstrations for all three environments (MetaWorld, RoboSuite, and Franka Kitchen). These experiments are conducted over an inpainted version of the original demonstrations, removing the robot body from all demonstration videos. The evaluation takes place with the robot body removed (keeping only the floating end-effector(s)).

Regarding criterion 3), as the human embodiment is strongly tied to the dataset, we apply a recent work (Chang et al., 2024) that performs agent inpainting, removing the human from each and every frame in the dataset. This synergizes with our approach to leveraging the hands as a substitute for objects, as the network is forced to rely on the perceived motion of the object to estimate transforms derived from the 3D hand predictions. This is opposed to if the hands were left in the video, in which case the network would over-rely on the movements of the hands in predicting motion, resulting in learned representations that fail to generalize towards robotic domains. We train our vision network over the inpainted dataset and freeze the learned representations for the learning of control policies.

## 4 EXPERIMENTS

In our experiments, we demonstrate the effectiveness of our contrastive learning objective in generalizing beyond the egocentric training dataset, serving as a useful frozen visual representation for motor policy learning. In Section 4.1 we describe the training and environment details of our policy learning evaluation. In Section 4.2 we first demonstrate that motion sensitive visual representations learned through our contrastive learning formulation enable better performance and sample efficiency over a variety of unseen environments and tasks. Second, in Section 4.3 we ablate the various components of our contrastive learning formulation, justifying the design choices of our proposed methodology. Finally, in Section 4.4 we show the learned representations are sensitive to motion not just over the EPIC Kitchens dataset, but over the out-of-domain videos taken from the RL environment.

## 4.1 Behavioral Cloning Evaluation

We perform experimentation over three prominent robot manipulation environments - Meta-World, RoboSuite, and Franka-Kitchen. Our experiments are done over a total of 21 robot manipulation tasks, simulated with the MuJoco physics engine, in accordance with the task distribution defined by (Hu et al., 2023). See Figure 4 for example tasks across the 3 environments.

We evaluate our visual representations using behavioral cloning, which has been reported to be a reliable algorithm for evaluating visual representations in policy learning (Hu et al., 2023). The visual representation backbone (ResNet50) is frozen during the training of the policy network, such that the behavioral cloning tasks and environments do not influence the training of the visual backbone. We vary the number of demonstrations the policy trains over and report the interquartile median (IQM) as it is less sensitive to outliers and provides reduced uncertainty. All results are reported over 4 independent runs, with different seeds.

Following the findings of (Hu et al., 2023), we opt out of performing experimentation where we evaluate our visual representations in the training of policies through reinforcement learning. There are known issues in evaluating visual representations for RL.

## 4.2 Comparisons

Here we evaluate the extent to which representations produced by our method outperform existing visual representations found to work well for policy learning in behavioral cloning settings. We take 4 methods as reported in (Hu et al., 2023) - that is, MoCov2 (Chen et al., 2020) (an unsupervised learning method using instance discrimination as a pre-training task), VICReg (Bardes et al., 2021) (a semi-supervised method that matches features within the same image based on pixel distance), VIP (Ma et al., 2022) (a similarly self-supervised representation learning method) and R3M (Nair et al., 2022) (a fully supervised representation learning method method utilizing linguistic labels during training).

| Environment | +Temp | +Act | +Mark | Ours |
|---|---|---|---|---|
| Robosuite | 39.75% | 45.91% | 48.73% | **49.12%** |
| Meta-World | 74.93% | 75.08% | **80.12%** | 79.90% |
| Kitchen | 53.51% | 57.19% | 58.01% | **58.31%** |
| All | 56.06% | 58.86% | 62.28% | **62.44%** |

Table 1: The component evaluations we perform. +Temp corresponds to stacking 3 visual representations across time during the policy training/evaluation stage, +Act corresponds to swapping motion targets for action label targets, +Mark is where we superimpose the marker(s) over EPIC Kitchens training videos and simulation environment. Ours is the entirety of the proposed framework.

We perform two sets of experiments. The first set of experiments is conducted over the original robot demonstrations and simulation environment. The second set of experiments is conducted over an inpainted version of the robot demonstrations, with the robot body removed from each frame, and the robot body is similarly removed from the simulation environment, keeping only the end effector in both instances.

## 4.3 Component Evaluations

Here we support the design choices of our contrastive learning framework through ablating components. Table 1 provides ablation results, and ablation descriptions are provided below:

**Temporal:** In this evaluation we explore the impact of additional information provided to our model: (+Temp) explores the extent to which temporal information benefits policy learning. We incorporate temporal information from the contrastive formulation by applying our backbone network independently over 3 frames, stacking the resultant feature vectors along the channels, and feeding the entire output to policy training.

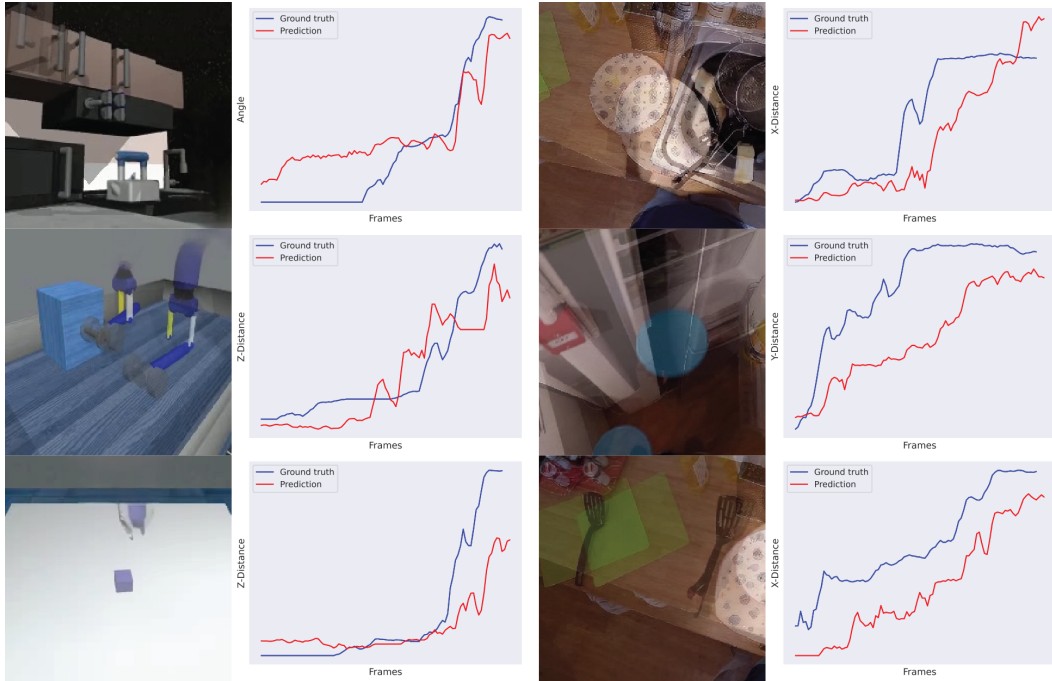

Figure 7: This figure illustrates the application of our motion prediction network across various manipulation tasks. For each subfigure we show an RGB image and a time-series plot depicting the predicted transforms for distinct actions: opening a door, moving the plate water, hitting a hammer, placing a container, picking and placing a block, and moving a spatula. Our method accurately captures the continuous progression of motion, demonstrating the sensitivity to motion of our learned representations. We demonstrate sensitivity both to rotational motion and translational motions.

**With Action labels:** This ablation explores the extent to which the improvement in performance due to our method is provided by the contrastive loss formulation. The contrastive loss formulation is ablated in +Act, where it is replaced by a classification loss across the action categories.

**+Mark** Here we superimpose red marker(s) over the EPIC Kitchens training videos where the hands are detected, along with a red marker where the robot end effector(s) is in both the robot demonstration videos and the simulation environment during evaluation. This in effect trivializes the learning of our vision backbone over the motion targets, as the network no longer is forced to attend to the object of interaction, and can instead attend to the marker for predicting the relative 3D transform between frames. See the right side of Figure 4 for the size and placement of the markers.

### 4.4 MOTION GENERALIZABILITY

Here we perform experimentation over a small subset of videos belonging to the EPIC Kitchens dataset and the simulation environments (see Figure 7). We compute the frame-to-frame motions using our vision network trained over EPIC Kitchens, plotting the predicted motions along the y-axis. These examples showcase instances where our network is capable of generalizing beyond EPIC Kitchens in capturing robot manipulation motions in simulation. We note while we did not perform real-world experimentation, it has been demonstrated empirically (Nair et al., 2022; Hu et al., 2023) over the models we compare against that improved success rates in simulation translate into the real-world.

### 5 DISCUSSION

We observe that the addition of our contrastive loss results in general improvements between the results of our method in Figure 6 and the results of state-of-the-art methods (MoCoV2, VICReg,

VIP and R3M) across both Figure 5 5 and Figure 6. Interestingly, we observe sizable performance gains across all 3 environments in our method when evaluated over the environment with the robot removed - we posit that this is because the motion in the scene is dominated by the arm and base of the robot body, which is less useful a cue than the motion belonging to the object. Removing the robot body results in a focused modeling of the end effector and object of interaction.

We also observe a drop in performance with the usage of action labels instead of our motion targets when evaluated on policy learning. This indicates that the reason for the observed performance improvements due to our contrastive learning method is *not* solely due to the usage of egocentric video data, and that our motion targets serve as a powerful alternative to action labels that are difficult to acquire at scale.

Curiously, we observe that the incorporation of temporal information (+Temp vs Ours in Table 1 ) provides no meaningful improvements. This is rather unintuitive as the visual representations are sensitive to frame-to-frame level motion, and so are particularly amenable to aggregation over time. We note that our findings are consistent with other experiments done (Hu et al., 2023) that aggregate visual representations over time using other vision backbone networks. One possible explanation for this is the lack of dynamics in the task definitions across all environments.

## 6 CONCLUSION

This paper presents a novel contrastive training framework that enhances motion encoding in visual representations for robotic manipulation tasks. By using a loss which focuses on motion and utilizing the EPIC Kitchens dataset, our approach significantly improves the alignment between visual models and the dynamic nature of robotic manipulation. After training over EPIC Kitchens, model evaluations on behavioral cloning show a improvement in success rate over state-of-the-art methods across a benchmark of 3 environments and 21 object manipulation tasks.

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
