# OpenReview forum: "From Appearance to Motion: Aligning Visual Representations for Robotic Manipulation"
_ICLR.cc/2025/Conference — ICLR 2025 Conference Withdrawn Submission_

### Official Review · Reviewer_9jPu · 2024-10-31

**Soundness:** 3
**Presentation:** 2
**Contribution:** 2
**Rating:** 3
**Confidence:** 4

**Summary:**

Unsupervised visual representations are typically learned from appearance cues rather than motion. This paper proposes a contrastive learning framework that leverages motion-based predictions to learn visual representations. The method is pre-trained on the EPIC Kitchens dataset and evaluated in a simulation environment, comparing its performance against other visual pre-training baselines such as MoCo, VICReg, R3M, and VIP.

**Strengths:**

1. Research on learning inverse dynamics as an unsupervised learning objective for visual representation learning remains limited;
2. The key contribution of this paper, in my view, lies in leveraging human hand motion combined with video pre-training for robotics applications, rather than solely focusing on learning from motion itself.

**Weaknesses:**

1. Learning inverse dynamics models from visual inputs has been explored in the past (i.e. [1,6]). It would be good to discuss these papers in the context of this paper.
2. There are a lot of works that learns forward dynamics as a pre-training task (i.e. world models [2,3,4,5]). However, it is unclear from this paper whether learning inverse dynamics is better.
3. The reviewer is not sure what is contrastive in the setup, since there’s no positive and negative samples throughout the paper.
4. Lack of ablations and experiments on a real robotics setup. Many recent works such as Implicit behavior cloning [7] and diffusion policy [8] can work with <50 demonstrations. It is not obvious that the visual pre-training actually helps with policy learning.
5. The paper does not report confidence interval and it is hard to tell if the model has lead to a improvement over baselines (which the paper claims in the abstract).

[1] Agrawal, Pulkit, Ashvin V. Nair, Pieter Abbeel, Jitendra Malik, and Sergey Levine. "Learning to poke by poking: Experiential learning of intuitive physics." Advances in neural information processing systems 29 (2016).[1] Agrawal, Pulkit, Ashvin V. Nair, Pieter Abbeel, Jitendra Malik, and Sergey Levine. "Learning to poke by poking: Experiential learning of intuitive physics." Advances in neural information processing systems 29 (2016).

[2] Wu, Philipp, Alejandro Escontrela, Danijar Hafner, Pieter Abbeel, and Ken Goldberg. "Daydreamer: World models for physical robot learning." In Conference on robot learning, pp. 2226-2240. PMLR, 2023.

[3] Mengjiao Yang, Yilun Du, Kamyar Ghasemipour, Jonathan Tompson, Dale Schuurmans, and Pieter Abbeel. Learning interactive real-world simulators. In NeurIPS, 2023.

[4] Yilun Du, Sherry Yang, Bo Dai, Hanjun Dai, Ofir Nachum, Josh Tenenbaum, Dale Schuurmans, and Pieter Abbeel. Learning universal policies via text-guided video generation. In NeurIPS, 2024a.

[5] Kevin Black, Mitsuhiko Nakamoto, Pranav Atreya, Homer Walke, Chelsea Finn, Aviral Kumar, and Sergey Levine. Zero-shot robotic manipulation with pretrained image-editing diffusion models. In NeurIPS, 2023.

[6] Brandfonbrener, David, Ofir Nachum, and Joan Bruna. "Inverse dynamics pretraining learns good representations for multitask imitation." Advances in Neural Information Processing Systems 36 (2024).

[7] Florence, Pete, Corey Lynch, Andy Zeng, Oscar A. Ramirez, Ayzaan Wahid, Laura Downs, Adrian Wong, Johnny Lee, Igor Mordatch, and Jonathan Tompson. "Implicit behavioral cloning." In Conference on Robot Learning, pp. 158-168. PMLR, 2022.

[8] Chi, Cheng, Zhenjia Xu, Siyuan Feng, Eric Cousineau, Yilun Du, Benjamin Burchfiel, Russ Tedrake, and Shuran Song. "Diffusion policy: Visuomotor policy learning via action diffusion." The International Journal of Robotics Research (2023): 02783649241273668.

**Questions:**

1. Taking a step towards the representation learning problem: lately we have seen more sample efficient behavior cloning papers (i.e. Implicit BC, Diffusion Policy) that works with 25-50 demonstrations. How well does these methods do on the environments? Is there a benefit from large scale visual pre-training
2. There are other works [1,2] that have been pre-trained on human manipulation datasets, but they do not extract hand pose. What about visual pre-training methods that has been applied on epic kitchen and other human datasets? I.e. MVP and VC-1?
3. I think there’s still a gap between the representations, as simulation environments may be largely different from real. Maybe you can try the inverse dynamics pre-training in the simulation (after pre-training on human hand data) and then train the policy.
4. How many trials are conducted for each of the simulation experiments?

[1] Radosavovic, Ilija, Tete Xiao, Stephen James, Pieter Abbeel, Jitendra Malik, and Trevor Darrell. "Real-world robot learning with masked visual pre-training." In Conference on Robot Learning, pp. 416-426. PMLR, 2023.

[2] Majumdar, Arjun, Karmesh Yadav, Sergio Arnaud, Jason Ma, Claire Chen, Sneha Silwal, Aryan Jain et al. "Where are we in the search for an artificial visual cortex for embodied intelligence?." Advances in Neural Information Processing Systems 36 (2023): 655-677.

---

### Official Review · Reviewer_Rdyw · 2024-11-02

**Soundness:** 1
**Presentation:** 1
**Contribution:** 1
**Rating:** 1
**Confidence:** 5

**Summary:**

In this paper, the authors claimed that current representation learning methods for robotic manipulation cannot model the internal motion dynamics of actions and proposed one framework that utilizes hand motions as a contrastive learning target. The experiments are conducted on MetaWorld, RoboSuite, and Franka Kitchen. However, under common settings, the proposed approach did not demonstrate any state-of-the-art performance, but the authors found their method effective when removing the robot body.

**Strengths:**

The authors conducted extensive experiments on three benchmarks although the unsatisfactory results.

**Weaknesses:**

- The claims that “… it is crucial to model motion since manipulation is fundamentally defined by motion rather than ‘appearance’” is far from the truth. Precise localization of manipulation objects is of vital importance to the success of an action. The misunderstanding leads the author to design the whole contrastive training approach in a counter-intuitive way, where manipulators in each image are even directly removed.
- Many commonly-used representation learning methods with better performances in the field of robotic manipulation are not compared with in your experiments, such as MVP, Voltron, VC-1, and MPI.
- The experiments under common settings cannot demonstrate the effectiveness of your approach. The advantages when removing the robot body from all demonstration videos are mainly because your methods have already removed human hands during pre-training.

**Questions:**

- Could you add more experiment results of the sota methods as I mentioned in Weaknesses under the normal settings?

---

### Official Review · Reviewer_6Ds2 · 2024-11-02

**Soundness:** 2
**Presentation:** 2
**Contribution:** 2
**Rating:** 5
**Confidence:** 4

**Summary:**

This paper proposes a contrastive learning framework aimed at improving the motion representation capabilities of pre-trained vision models for robotic manipulation tasks. Recognizing a misalignment in standard pre-trained models that focus on appearance rather than motion, the authors introduce a contrastive objective that emphasizes motion, trained on the EPIC Kitchens dataset. Experimental results in behavioral cloning across several environments and benchmarks demonstrated some improvement.

**Strengths:**

- The paper identifies an important problem in current visual models used in robot learning: their reliance on appearance-based representations, which often misalign with the requirements of manipulation tasks that are inherently motion-centric.

- The proposed contrastive loss formulation makes sense because it learns to emphasize motion over appearance, making it well-suited for robotic applications.

- Pre-training dataset, EPIC, which is egocentric, offers an effective strategy for capturing hand-object interactions, making it a good prior for robotic manipulation scenarios.

**Weaknesses:**

- Motivation is lacking. Not because the motivation itself is weak, but because the related work leading to your conclusion that appearance-based representations misalign is missing. A more comprehensive RW is needed.
- Technical details are hand-wavy: The paper lacks mathematical rigor - contrastive loss, etc. was verbally communicated and was not defined well using mathematical notations. Actually this makes the paper look incomplete and rushed.
- Experiments are insufficient: the paper does not sufficiently explore how this contrastive framework differs from other similar self-supervised methods (e.g., VICReg, R3M). The idea itself is not new. What makes your approach different? Explicit comparisons or theoretical rationale differentiating this framework from other contrastive or self-supervised methods would improve clarity.

**Questions:**

- Add specific mathematical formulations of the contrastive loss and provide pseudo-code or a more detailed description of the training steps.
- Provide a more thorough comparison to related methods, especially those using contrastive learning and self-supervised techniques.
- More literature review.

---

### Official Review · Reviewer_mGrN · 2024-11-04

**Soundness:** 2
**Presentation:** 2
**Contribution:** 2
**Rating:** 3
**Confidence:** 4

**Summary:**

The paper proposes an approach for visual pre-training for robotic manipulation. The approach falls into the broad the category of methods that pre-train a vision encoder on non-robot images/videos and use the pre-trained and frozen visual representations for downstream policy learning.

Given two video frames, the proposed approach estimates hand poses using an off-the-shelf hand pose estimator and computes a transformation between them. It then trains a vision encoder to predict the estimated transformation using a contrastive loss. Additionally, the vision encoder operates on images with hand pixels removed/inpainted.

The vision model is pre-trained on the Epic Kitchens dataset and evaluated in simluation. The results show comparable or better performance than several vision encoders from prior work.

**Strengths:**

- Using motion cues to pre-train vision encoders for robotic manipulation is a promising direction

**Weaknesses:**

- The approach relies on off-the-shelf models to construct training targets and model inputs which is a bit complex and would make extending this approach to larger video collections more challenging
- The empirical results are overall limited. The approach is evaluated on relatively simple simulation environments which makes it difficult to draw robust conclusions on the performance

**Questions:**

- 1) How does the approach compare to stronger vision encoders trained from appearance alone? Like those based on reconstruction Ilija Radosavovic, Tete Xiao, Stephen James, Pieter Abbeel, Jitendra Malik, and Trevor Darrell. Real-world robot learning with masked visual pre-training. In Conference on Robot Learning, pp. 416–426. PMLR, 2023.

- 2) How are the transformations between frames computed exactly? My understanding is that the estimated hand poses are in the view space rather than the world space. It would be good to describe the procedure

- 3) What is the effect of in-painting on downstream performance? It would be good to ablate this
are transformation computed exactly

- 4) What is the amount of data that the model is trained on? All of Epic kitchens? A curated subset of Epic Kitchens? What was the curation procedure used if any? It would be good to describe this

---

### Note · Authors · 2024-11-26

I have read and agree with the venue's withdrawal policy on behalf of myself and my co-authors.